# Nano-Structure Evolution and Mechanical Properties of Al_x_CoCrFeNi_2.1_ (x = 0, 0.3, 0.7, 1.0, 1.3) High-Entropy Alloy Prepared by Mechanical Alloying and Spark Plasma Sintering

**DOI:** 10.3390/nano14070641

**Published:** 2024-04-07

**Authors:** Guiqun Liu, Ziteng Lu, Xiaoli Zhang

**Affiliations:** School of Materials Science and Technology, North Minzu University, Yinchuan 750021, China; gqliu10b@alum.imr.ac.cn (G.L.);

**Keywords:** Al_x_CoCrFeNi_2.1_ HEAs, spark plasma sintering, nano-structure evolution, mechanical properties, mechanical alloying

## Abstract

The Al_x_CoCrFeNi_2.1_ (x = 0, 0.3, 0.7, 1.0, 1.3) multi-component high-entropy alloy (HEA) was synthesized by mechanical alloying (MA) and Spark Plasma Sintering (SPS), The impact of the percentage of Al on crystal structure transition, microstructure evolution and mechanical properties were studied. Crystal structure was investigated by X-ray Diffraction (XRD) and Scanning Electron Microscopy (SEM). The results show that with the increasing of Al content, the crystal structure of the alloys gradually transformed from a nanocrystalline phase of FCC to a mix of FCC and BCC nanocrystalline. The hardness was found to increase steadily from 433 HV to 565 HV due to the increase in fraction of BCC nanocrystalline phase. Thus, the compressive fracture strength increased from 1702 MPa to 2333 MPa; in contrast, the fracture strain decreased from 39.8% to 15.6%.

## 1. Introduction

The design concept of traditional alloys is usually based on one or two alloying elements as the main component, and sometimes a small number of other elements will be added to enhance the performance, the content of alloying elements is often between 1–5%, and in a few cases it can reach 10%, such as steel [1]. HEAs usually consist of at least five main components, each with content varying from 5% to 35% at. [2,3,4,5,6,7]. Traditional alloy theory holds that with the gradual increase in alloying elements, complex phases or metal components are easy to form in multi-principal alloy systems, resulting in poor mechanical properties of alloys [8]. The high mixing entropy of HEAs can lead to nano-lattice distortion and hysteresis diffusion, so HEAs usually have simple solid solutions and amorphous structures, rather than intermetallic compound [3].

In the early research stage of high-entropy alloys, scholars mainly studied the development and design of single-phase high-entropy alloys. For example, FeMnCoCrNi HEA with only FCC nanocrystalline structure has high plasticity but low strength [9], while V_20_Nb_20_Mo_20_Ta_20_W_20_ with only BCC nanocrystalline structure has high strength but low plasticity [10]. In 2014, the concept of eutectic high-entropy alloy (EHEA) was first proposed [11]. It takes into account that the advantages of high strength of BCC nanocrystalline phase while high ductility of FCC nanocrystalline phase. This feature is well balanced in the studies of some scholars [12,13]. However, there are few reports on the synthesis of eutectic alloys by mechanical aggregation- Spark Plasma Sintering. In HEA, the content of Al element with larger atomic radius is usually changed to explore the effect on the structure and properties of HEA. In this study, the effect of Al content in Al_x_CoCrFeNi_2.1_ on the microstructure evolution and mechanical properties of the alloys will be discussed.

## 2. Experimental Details

Powders of Al, Co, Cr, Fe, and Ni with purity more than 99.5% were used as a starting material for mechanical alloying in a FP-2000 planetary ball mill (FOCUCY, Changsha, China) with tungsten carbide (WC) vials and balls. Powders were milled in dry condition under the protective argon (Ar) atmosphere up to 20 h, followed by anhydrous ethanol for 4 h under wet conditions. In the whole process, the speed was 400 rpm with a ball to powder ratio (BPR) of 10:1 (Figure 1a). The milling operation adopted intermittent operation, running for 30 min and stopping for 30 min, in order to prevent excessive temperature rise and protect the powder from oxidation in the bottle. At the initial milling stage (0 h) and 1 h, 3 h, 6 h, 10 h, 20 h, 24 h, the powder was removed and characterized by XRD and SEM. After milling, the samples (Φ20 mm in diameter and 5 mm in thickness) compressed by alloyed powder were put into a vacuum drying oven to dry at 80 °C for 48 h, and then the samples was consolidated at 1100 °C and 40 MPa for 8 min in SPS furnace (SPS-20T-10, Shanghai Chenhua Technology Co., Ltd., Shanghai, China), and then cooled to room temperature with the furnace (Figure 1).

The crystal structure of the as-preserved alloy was characterized by X-ray diffractometer (XRD, Shimadzu, XRD-6000, Kyoto, Japan) with Cu Kα radiation. The microstructure of the powders was observed using SEM (ZEISS, SIGMA 500, Darmstadt, Germany), the chemical composition was analyzed using energy-dispersive X-ray spectroscopy (EDS). Bulk hardness of the sectioned and polished specimens was measured using a Vikers hardness tester with a load of 9.8 N and dwell time of 10 s. Before the hardness testing, it was necessary to cut 1 mm off the surface of the alloy block with an electric spark cutting machine cutter to prevent the influence of C atom penetration on the hardness of the alloy during consolidation. The samples for the compressive stress–strain test were cut into a cylinder of Φ3 mm × 6 mm by a wire-cutting machine. Compressive stress–strain tests at room temperature were conducted by an MTS-E45.105 Electronic universal testing machine with a static compression rate of 1.0 × 10^−3^·S^−1^. For the convenience of description, the alloys CoCrFeNi_2.1_, Al_0.3_CoCrFeNi_2.1_, Al_0.7_CoCrFeNi_2.1_, AlCoCrFeNi_2.1_, and Al_1.3_CoCrFeNi_2.1_ were named Al_00_, Al_03_, Al_07_, Al_10_, and Al_13_, respectively.

## 3. Results and Discussion

### 3.1. Criterion of Solid Solutions of HEAs

In order to better predict the solid solution phase and stability of HEAs, previous scholars have introduced some theories, including atomic size difference (*δ*), enthalpy of mixing (Δ*H_mix_*), entropy of mixing (Δ*S_mix_*) [14], valence electron concentration (VEC) [15], and the combined effect of enthalpy of mixing, entropy of mixing, and melting temperature (Ω) of alloy system.

The atomic size difference (δ) is calculated via Equation (1)
(1)$$\delta =100\sqrt{\sum_{i=1}^{n}c_{i}{\left(1-r_{i}/\bar{r}\right)}^{2}}$$

In the equation, *c_i_* is the atomic percentage of the *i* th principal element, *r_i_* is the atomic radius of the *i* th principal element, and $\bar{r}=\sum_{i=1}^{n}c_{i}r_{i}$ is the average atomic radius.

Mixing enthalpy Δ*H_mix_* is calculated as follows:(2)$$\Delta H_{mix}=\sum_{i=1,i\neq j}^{n}\Omega_{ij}c_{i}c_{j}$$

In the equation, $\Omega_{ij}=4\Delta H_{ij}^{mix}$,$4\Delta H_{ij}^{mix}$ is the enthalpy of mixing when the *i* and *j* principal elements form a binary liquid alloy.

And mixing entropy (Δ*S_mix_*) can be represented by Equation (3)
(3)$$\Delta S_{mix}=-R\sum_{i=1}^{n}c_{i}lnc_{i}$$
where *R* is the gas constant.

When Zhang Yong et al. were studying the formation law of HEAs phase [14], they put forward the Ω criterion considering the mixing enthalpy and melting point of the alloy. This equation is as follows:(4)$$\Omega =\frac{T_{m}\Delta S_{mix}}{\left|\Delta H_{mix}\right|}$$

*T_m_* is the theoretical melting point of the alloy, the equation is calculated as (5)
(5)$$T_{m}=\sum_{i=1}^{n}c_{i}{\left(T_{m}\right)}_{i}$$
where *i* represents the melting point of the *i*-th element in the alloy period, and in addition to the above thermodynamic criteria, electronic structure such as *VEC* (valence electron concentration) is another parameter that helps to predict the HEAS phase. *VEC* is defined by Equation (6)
(6)$$VEC=\sum_{i=1}^{n}c_{i}{\left(VEC\right)}_{i}$$
where *(VEC)_i_* is the *VEC* for the *i*-th element. The mixing enthalpy of each element pair in the alloy and the characteristic parameters of each element are shown in Table 1 and Table 2, respectively.

### 3.2. The Grain Is Refined by Ball Milling

Al_x_CoCrFeNi_2.1_ HEAs were produced via the MA process and samples were examined using the XRD test after different MA period times, which are presented in Figure 2. In the initial period of ball milling (0 h), the characteristic peaks of each element of the High-Entropy Alloy (Al, Co, Cr, Fe, Ni) can be clearly identified. After milling for one hour, the diffraction peak intensity of each element begins to decay, and after milling for three hours, the characteristic peak corresponding to Al in Figure 2b–e disappears firstly. As we know, from a chemical point of view, the chemical bond of a substance with low energy leads to a lower melting point of the substance, so the substance has a higher diffusion rate [17]. Therefore, there are five elements in this system, and the XRD characteristic peak of Al disappears first. After milling for 6 h, the second strongest peak of XRD (51.8°) corresponded to element Co as seen in Figure 2a–e, because the first strongest peak of Co (44.5°) overlaps with other component characteristic peaks. Therefore, the disappearance of the 51.8° characteristic peak determines that the Co diffuses into other component nanolattices. The characteristic peak strength of Fe, Ni, and Cr is continuously weakened, and the peak width is continuously widened, which indicates that the nanostructure grain is refined during the milling process. After 20 h, the powder was alloyed. And anhydrous ethanol was added as the process control agent for further milling for 4 h. The XRD pattern showed that the peak height and peak width were further increased, because the alloy powder would be milled finer after adding alcohol wet grinding. It is worth noting that the alloy powder with Al content x = 0 or 0.3, formed FCC solid solution single-phase high-entropy alloy after alloying, Al = 0.7, 1, 1.3 formed FCC nano solid solution as the main phase, BCC nano solid solution as the secondary phase of the biphasic high-entropy alloy. It can be seen that with the increase in Al content, the proportion of FCC solid solution in the alloy will decrease, which is conducive to the formation of BCC solid solution. In general, the order in which the XRD diffraction peaks of each element in the high-entropy alloy in this study disappear is as follows: Al→Co→Ni→Fe→Cr. This is confirmed by other scholars [17]; that is, when a high-entropy alloy is in ball milling, the alloying rate of each element is negatively correlated with its melting point. The higher the melting point, the more difficult it is to complete alloying. In addition, it is also related to the plasticity of the alloy, that is, the worse the plasticity when the melting points are close to each other, the easier it is to achieve alloying. The powder of each element needs to undergo continuous crushing, agglomeration, and cold welding, so in this process, the less plastic elements are easier to diffuse into the new nano-crystal structure [17].

Table 3 shows grain size calculated by Scherrer formula and XRD pattern of alloy powder after 24 h ball milling. It can be seen by calculation that after 24 h milling of Al_00_–Al_13_, the grain size of nano FCC phase and nano BCC phase in the alloy powder has become very small, reaching the nanometer level (3–20 nm), Compared with other reports, the alloying time and grain size are greatly shortened and refined [18]. A large number of nanocrystals are introduced by MA, which is helpful to improve the properties of the alloy.

SEM morphology of AlCoCrFeNi_2.1_ HEAs milled for different time periods are presented in Figure 3. As shown in the Figure 3a, the unique morphology of the powder of each metal is clearly visible in the initial stage of the process. In this period, the powder particles are soft and have a certain ductility, so they are easy to deform and the cold-welding rate is high. In general, deformation and cold welding are the two main mechanisms for controlling nano-particle size during short milling periods [19,20,21], with an increase in milling time, The metal nano-particles will be aggregated into larger particles, and then broken and aggregated to form particles of different sizes, and then added anhydrous ethanol for wet grinding in the last 4 h to form lamellated particles with a size of no more than 100 μm. EDS analysis was performed on region A in Figure 3g, and the results were shown in Table 4. It can be seen that only the content of Al in the obtained alloy is different to that of the nominal component. This is probably because the relative atomic mass of the Al is lower, and some of the lamellar powder particles in the selected region are not flat, so there is some error.

### 3.3. SPS Synthesis of Biphasic High-Entropy Alloys

Figure 4 shows the XRD pattern of the alloy after SPS. Combined with Figure 2, it can be seen that only a single FCC nano-phase appears in the alloys Al_00_ and Al_03_ after milling for 24 h, and no phase transition occurs after SPS. The BCC nano-phase content in the samples of Al_07_, Al_10_, and Al_13_ after 24 h MA was higher than that of FCC phase, as seen in Figure 2. This indicates that the partially metastable BCC solid solution changes to the more stable FCC solid solution after SPS [22]. In Figure 4**,** except for the FCC and BCC nano-phases, we also observed a peak corresponding to Cr_7_C_3_, which was the same as in the previous literature on the synthesis of HEAs from MA-SPS, because Cr_7_C_3_ was formed in the process of MA-SPS due to the high affinity between Cr and C in the alcohol added at the last stage of milling [23]. The phase fraction of BCC in FCC is calculated by Equation (7), and the results are shown in Table 5
(7)$$V_{BCC}=\frac{V_{BCC}}{V_{BCC}+V_{FCC}}$$

In Equation (7), *V_FCC_* and *V_BCC_* are, respectively, the ratio of peak area of FCC and BCC, as seen in Figure 4.

Figure 5 shows the microstructure image of Al_x_CoCrFeNi_2.1_ (x = 0, 0.3, 0.7, 1.0, 1.3) alloy powder after SPS. The lower left corner is the enlarged image of the local area. From the five SEM images in Figure 5, there are almost no pores, which proves the high density of the alloy prepared by MA-SPS process. Table 6 shows the EDS analysis results of Spot 1–15 in SEM images of high-entropy alloys with different contents of Al.

When Al = 0.0, the microstructure was composed of a light-colored matrix, brown irregular region, and black nearly circular region. EDS analysis was performed on Spot1 and Spot2 in Figure 5a, and the results were shown in Table 6, where the content of each element in Spot 2 region was close to the nominal component, indicating that the composition of SPS was uniform after sintering. There is a higher Cr content than the nominal component near Spot 1, and C element is also detected, so it is determined that this phase is Cr_7_C_3_, which is consistent with the XRD pattern of the alloy in Figure 4. Near Spot 3, there is a high content of Al and O. The reason for the presence of Al_2_O_3_ in Al = 0.0 is the residual Al in cold welding on the ball mill tank during the previous experiment. In this experiment, due to cold welding and diffusion, a small amount of Al_2_O_3_ appears after the completion of sintering. Figure 5b shows the SEM image when Al = 0.3, and the microstructure is similar to that of Al = 0.0. As can be seen from Table 6, the composition fluctuation of Al = 0.3 alloy is relatively large, which is caused by more serious lattice distortion caused by the larger atomic radius of Al, which is not conducive to the diffusion of other atoms. Figure 5c shows the SEM image when Al = 0.7, and its microstructure is composed of a light-colored matrix, brown irregular region, and black circular region. EDS analysis near Spot 7, 8, and 9 showed that the high content of Al and Ni elements near Spot 7 is due to the fact that Al and Ni have the most negative atomic pair mixing enthalpy. In short, when a solid solution begins to appear between elements due to the high-speed impact of the grinding ball during the milling process, Ni atoms are more likely to occupy the position of Al atoms, so it is easy for Al and Ni to combine. In addition, it is found that Co is slightly more than the nominal component in the brown region, because mixing Co and Al also have the second-lowest mixing enthalpy, the only one lower than that of mixing Al and Ni, and the mixing enthalpy of Co and Ni is 0, as seen in Table 1, which leads to more Co in the BCC phase. The content of Al and O was higher near Spot 8. The Fe, Co, and Cr contents near Spot 10 were higher than the nominal components. According to the studies of other scholars [23] and EDS results, it is determined that the brown phase is the BCC solid solution phase rich in Al-Ni, and the light-colored matrix is the FCC solid solution phase rich in Fe, Co, and Ni. The microstructure of Al = 1.0 and 1.3 is shown in Figure 5c, d. According to the EDS results of Table 4, the brown phase is the Al-Ni phase and the light-color phase is the Fe-Co-Cr phase. Al = 1.0 microstructure shows that BCC phases are point-like and clustered in an irregular shape and FCC phases are cross-distributed. In the microstructure of the Al = 1.3 alloy, the BCC phase is irregular and densely distributed on the FCC phase.

In summary, combined with Figure 4 and Figure 5, it can be concluded that with the increase in Al content, the single FCC solid solution phase in Al_x_CoCrFeNi_2.1_ high-entropy alloy is gradually transformed into a biphasic high-entropy alloy in which FCC and BCC solid solutions coexist, and the nano BCC phase proportion is gradually increasing. The FCC solid solution phase is rich in Fe-Co-Cr and the BCC phase is rich in Al-Ni.

As mentioned above, the Al_x_CoCrFeNi_2.1_ high-entropy alloy prepared in this paper mainly consists of nano FCC phase or nano BCC + FCC phase. It was previously thought that a simple solid solution can be formed when$\;\Delta S_{mix}\geq 1.5R=12.47{\;\mathrm{JK}}^{-1}{\mathrm{mol}}^{-1}$, but a large number of studies and the study in this paper show that the mixing entropy criterion is not the only criterion. According to the atomic radius difference criterion proposed by Yang et al. [24], when δ is less than or equal to 6.6% and Ω is greater than or equal to 1.1, solid solutions are easily formed in high-entropy alloys. Guo et al. [15] proposed that when VEC ≥ 8, a single solid solution is formed; 6.87 ≤ VEC < 8 is a solid solution with FCC + BCC biphasic structure; and when VEC < 6.87, a single BCC solid solution is formed. The calculation results according to the Formulas (1)–(6) are shown in Table 7. According to the calculation, when Al = 0.0, the criterion of ∆*S_mix_* is not satisfied, but Ω and VEC meet the conditions to form FCC structure solid solution, and finally form single phase FCC phase when Al = 0.0. When the Al content increased to 0.7, VEC < 8 was consistent with the judgment of XRD and SEM images, and BCC phase appeared. When Al = 1.0 and 1.3, Ω, VEC and ∆*S_mix_* all meet the conditions of forming BCC + FCC biphasic solid solution, which is consistent with the experimental results.

### 3.4. Compression Properties and Fracture Morphology of Alloys

In order to investigate the mechanical properties of the AlxCoCrFeNi_2.1_ alloy undergoing in MA-SPS, its compression properties were tested at room temperature. The room temperature compressive stress–strain curves of the five samples in this experiment are shown in Figure 6. In HEAs, it is generally considered that the nano FCC phase has good plasticity and the nano BCC phase has high strength. The strength of the alloy material increases with the increase in BCC phase, and the strain also decreases. From Figure 6 and Table 8, we can see that the ductility of Al_00_ alloy is the best, reaching 39.8%, but its strength is only 1701 MPa., when Al = 1.3, the highest fracture strength is 2333 MPa and the fracture strain is only 15.6%. It can be seen in Figure 6 as the proportion of Al content increases, the compressive stress–strain curve of Al_x_CoCrFeNi_2.1_ HEAs shows a trend of increasing strength and decreasing strain, which is consistent with the traditional theory that BCC increases when the Al content increases. The nano BCC phase has high strength but poor plasticity, so the change trend of the alloy is that with the increase in Al content, the strength increases and the plasticity decreases.

Figure 7 shows the fracture morphology of Al_x_CoCrFeNi_2.1_ HEA after compression test at room temperature; the high-magnification image is placed in the left bottom. The fracture mode of Al_00_ and Al_03_ alloys is slip fracture. In the lower right corner of Figure 7a, the locally enlarged image is observed, and dimpling is observed in the red circle. In the fracture image of the Al_07_ alloy, in addition to the large area of slip fracture, intergranular fracture mode is observed in the blue circular pattern. The XRD diffraction pattern shown in Figure 5 above proves that a small amount of BCC phase appears, so the fracture mode of the Al_07_ alloy is different from that of Al_00_ and Al_03_, which is a fracture mechanism combining slip fracture and intergranular fracture. The fracture mode of Al_10_ is the same as that of Al_07_, but the intergranular fracture area of Al_10_ is significantly increased. Combined with the high-magnification SEM picture in Figure 7c, it can be observed that the distribution of BCC nano phase in Al_07_ alloy is relatively uniform, while the distribution of BCC and FCC nano phase in Al_10_ alloy is not uniformly interwoven, which may be the reason for the high hardness and yield strength of Al_07_. In the fracture image of Al_13_, there is almost no form of slip fracture observed, only a large number of intergranular fractures, which is also the reason for the worse strain of Al_13_, and the increase in BCC phase is also the reason for the increase in its strength.

## 4. Conclusions

In this article, mechanical alloying and spark plasma sintering were used to prepare high-entropy alloy. the alloy prepared by this method has a biphasic high-entropy alloy and showed excellent mechanical properties. The main conclusions are as follows:For the first time, the effect of Al content on AlCoCrFeNi_2.1_ HEAs was investigated by the combined MA-SPS technique. When Al = 0.0, 0.3, the alloy is composed of FCC nano phase, and when Al = 0.7, 1.0, 1.3, it consists of FCC-CoCrFe and BCC-AlNi nano phases.In terms of microstructure, Al = 0.0 and Al = 0.3 alloys exhibit a light-colored FCC nanocrystalline structure. The microstructure of Al = 0.7 shows that the light-colored FCC phase is the matrix, and the brown BCC phase appears in an irregular form. Al = 1.0 microstructure shows that BCC phases are point-like and clustered in an irregular shape and FCC phases are cross-distributed. In the microstructure of the Al = 1.3 alloy, the BCC nano phase is irregular and densely distributed in the FCC nano phase.The HEA prepared by MA-SPS has high hardness and strength. In this paper, with the increase in Al content, the hardness and strength of the alloy gradually increase, and the strain gradually decreases. When Al = 0.0, the maximum compressive strain is 39.8%, and the fracture strength and hardness are the lowest, which are 1701 MPa and 433 HV, respectively. When Al = 1.3, the highest fracture strength and hardness are 2333 MPa and 563 HV, but the compressive strain is only 15.6%.

## Figures and Tables

**Figure 1 nanomaterials-14-00641-f001:**
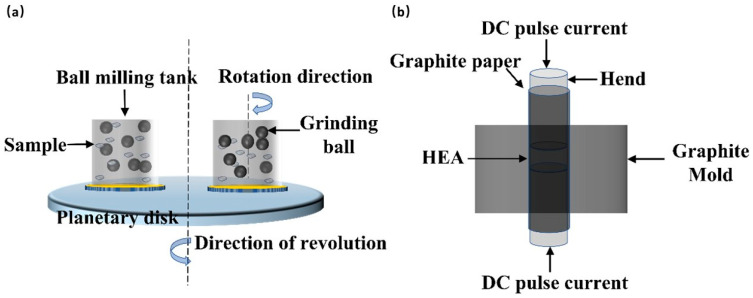
(**a**) Diagram of mechanized alloying process, (**b**) SPS diagram.

**Figure 2 nanomaterials-14-00641-f002:**
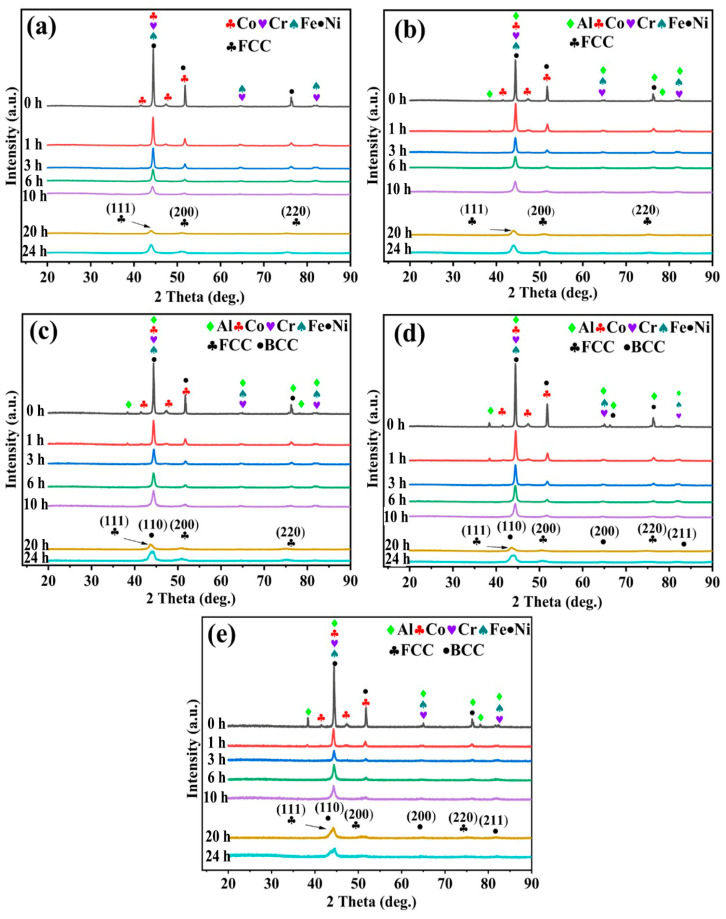
XRD patterns of the mechanically alloyed Al_x_CoCrFeNi_2.1_ HEAs after different time periods. (**a**) x = 0, (**b**) x = 0.3, (**c**) x = 0.7, (**d**) x = 1.0, (**e**) x = 1.3.

**Figure 3 nanomaterials-14-00641-f003:**
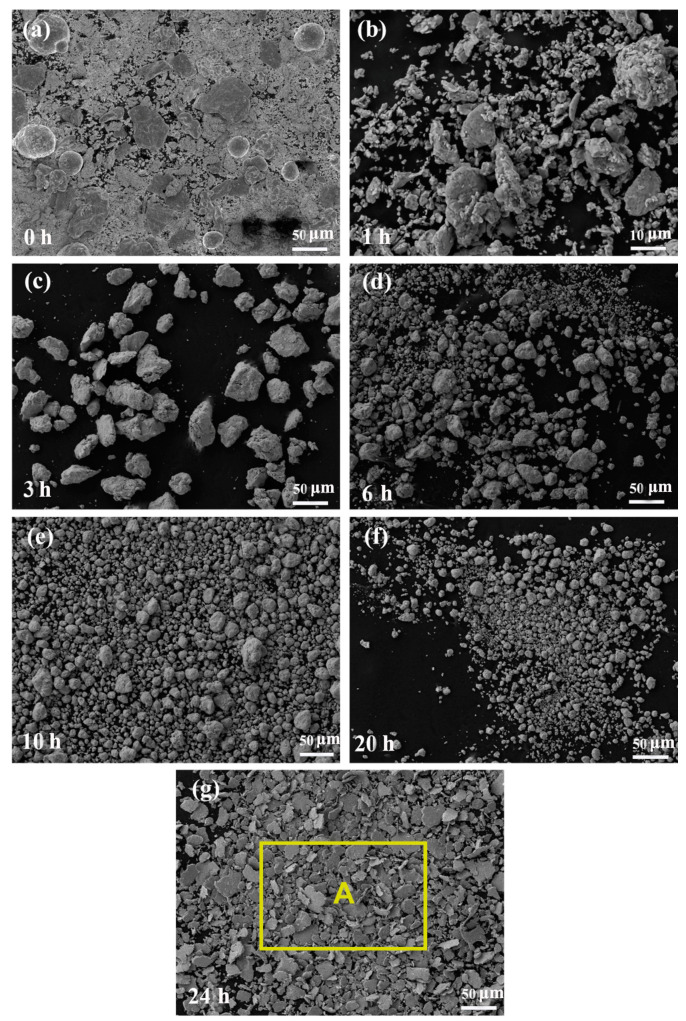
SEM microstructure of AlCoCrFeNi_2.1_ HEA after (**a**) 0 h, (**b**) 1 h, (**c**) 3 h, (**d**) 6 h, (**e**) 10 h, (**f**) 20 h, (**g**) 24 h of milling. EDS analysis was performed on region A in Figure 3g, and the results were shown in Table 4.

**Figure 4 nanomaterials-14-00641-f004:**
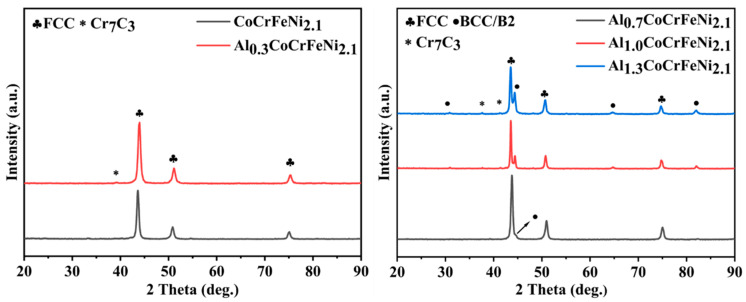
XRD pattern of sintered sample.

**Figure 5 nanomaterials-14-00641-f005:**
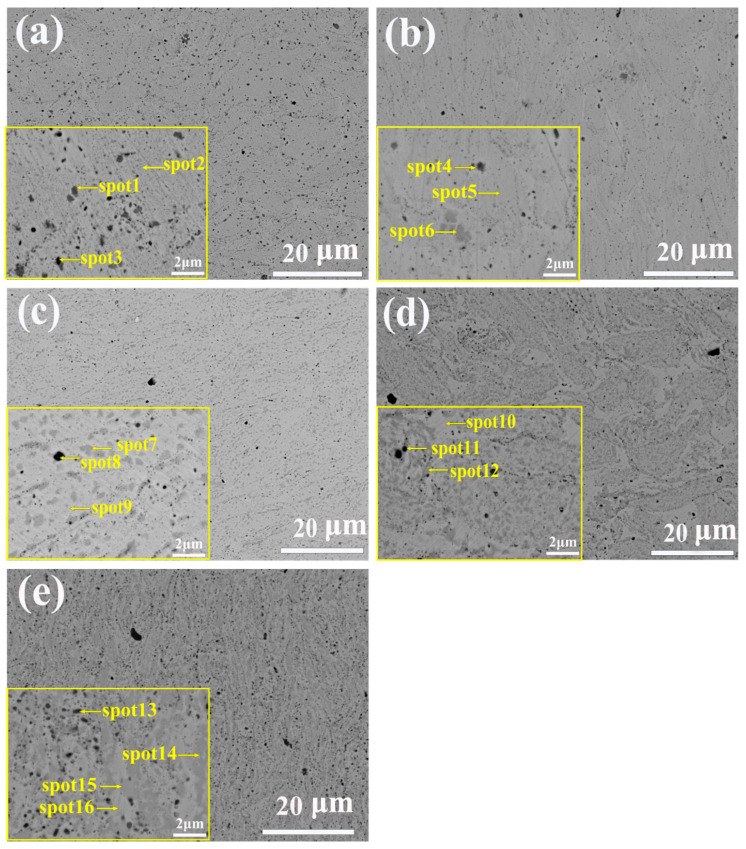
Microstructure of Al_x_CoCrFeNi_2.1_ image: (**a**) Al = 0.0, (**b**) Al = 0.3, (**c**) Al = 0.7, (**d**) Al = 1.0, (**e**) Al = 1.3.

**Figure 6 nanomaterials-14-00641-f006:**
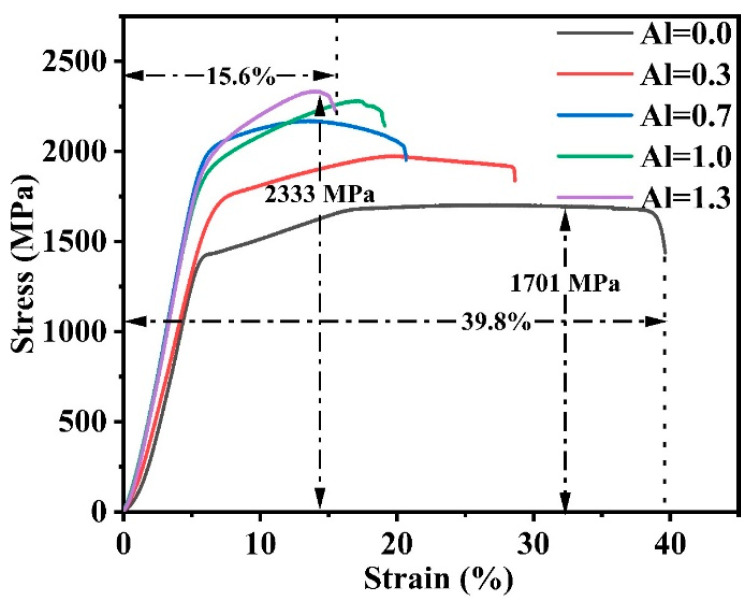
Compressive stress–strain curves of Al_x_CoCrFeNi_2.1_ HEAs samples at room temperature.

**Figure 7 nanomaterials-14-00641-f007:**
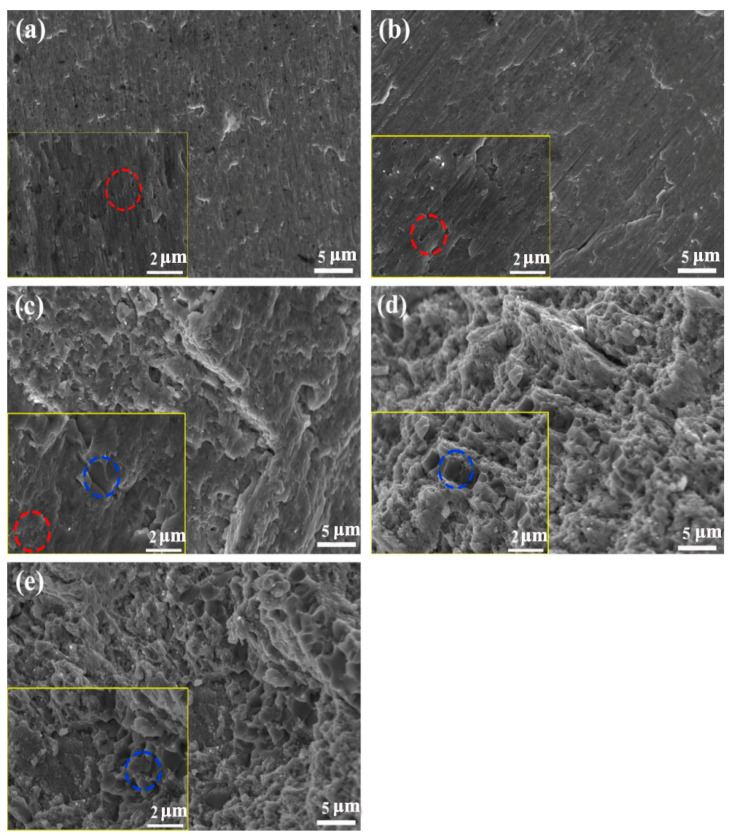
The fracture morphology of Al_x_CoCrFeNi_2.1_ (**a**) Al_00_, (**b**) Al_03_, (**c**) Al_07_, (**d**) Al_10_, (**e**) Al_13_.

**Table 1 nanomaterials-14-00641-t001:** Mixing enthalpy of atomic pairs in Al_x_CoCrFeNi_2.1_ (kJ/mol) [14].

Element	Al	Co	Cr	Fe	Ni
Al	0	−19	−10	−11	−22
Co		0	−4	−1	0
Cr			0	−1	−7
Fe				0	−2
Ni					0

**Table 2 nanomaterials-14-00641-t002:** Characteristic parameters of each element [16].

Element	Atomic Number	Crystal Structure	Atomic Radius (Å)	Fusing Point (°C)	Electron-Egativity (Pauling)	(VEC)	Molar Mass
Al	13	FCC	1.432	660	1.61	3	26.98
Cr	24	BCC	1.249	1857	1.66	6	51.99
Fe	26	BCC, FCC	1.241	1535	1.83	8	55.85
Co	27	HCP, FCC	1.251	1495	1.88	9	58.93
Ni	28	FCC	1.246	1453	1.91	10	58.69

**Table 3 nanomaterials-14-00641-t003:** Grain size of the alloy powder after 24 h milling.

Alloys	Crystalline Size (nm)	
	FCC	BCC
Al_00_	7.5	-
Al_03_	6.5	-
Al_07_	5.7	20.5
Al_10_	3.5	8.8
Al_13_	6.1	15.2

**Table 4 nanomaterials-14-00641-t004:** EDS results of AlCoCrFeNi_2.1_ alloy powder after 24 h ball milling (at. %).

Area	Al	Co	Cr	Fe	Ni
Nominal	16.39	16.39	16.39	16.39	34.42
HEAs	13.43	16.95	16.45	17.48	35.69

**Table 5 nanomaterials-14-00641-t005:** Proportion of FCC and BCC phases in AlxCoCrFeNi _2.1_ HEAs after SPS.

Alloys	FCC (%)	BCC (%)
Al_00_	100	-
Al_03_	100	-
Al_07_	93.5	6.5
Al_10_	71.2	28.8
Al_13_	67.8	32.2

**Table 6 nanomaterials-14-00641-t006:** Al_x_CoCrFeNi_2.1_ HEAs element content (in at. %) of the phases in Figure 5 according to the EDS spectrum.

Alloys	Region	Al	Co	Cr	Fe	Ni	C	O
Al_00_	Nominal	-	19.6	19.6	19.6	41.2	-	-
	Spot1	-	7.2	31.4	7.9	13.7	39.8	-
	Spot2	-	18.5	19.7	20.1	41.8	-	-
	Spot3	11.1	10.9	18.3	11.6	22.4	-	25.7
Al_03_	Nominal	5.5	18.5	18.5	18.5	39.0	-	-
	Spot4	12.8	13.8	14.2	13.8	28.7	-	16.6
	Spot5	6.5	18.7	17.3	18.6	38.9	-	-
	Spot6	3.9	4.1	43.9	5.0	5.6	37.4	-
Al_07_	Nominal	12.0	17.2	17.2	17.2	36.2	-	-
	Spot7	25.5	11.1	9.1	11.2	43.1	-	-
	Spot8	29.2	5.3	6.1	12.8	10.2	-	36.4
	Spot9	10.2	19.6	18.4	20.3	34.5	-	-
Al_10_	Nominal	16.4	16.4	16.4	16.4	34.4	-	-
	Spot10	8.9	18.5	18.5	19.9	33.9	-	-
	Spot11	33.8	4.9	5.6	5.0	10.9	-	39.9
	Spot12	21.6	13.7	12.9	13.5	38.3	-	-
Al_13_	Nominal	20.4	15.6	15.6	15.6	32.8	-	-
	Spot13	26.2	9.4	9.5	10.0	14.35	-	30.6
	Spot14	31.12	10.85	6.3	9.8	41.9	-	-
	Spot15	8.75	20.0	19.1	20.6	31.6	-	-
	Spot16	10.9	5.6	30.1	6.6	15.4	-	-

**Table 7 nanomaterials-14-00641-t007:** The values of parameters, Δ*H_mix_*, Δ*S_mix_*, δ, Ω, and VEC for the Al_x_CoCrFeNi_2.1_ alloy.

Alloys	$\Delta H_{mix}$ (J/mol K)	$\Delta S_{mix}\;(\mathrm{KJ}/\mathrm{mol})$	δ (%)	Ω	VEC
Al_00_	−3.83	11.00	1.02	4.47	8.63
Al_03_	−6.97	12.17	3.22	2.63	8.32
Al_07_	−10.14	12.73	4.43	1.82	7.95
Al_10_	−11.51	12.90	4.97	1.58	7.70
Al_13_	−13.38	12.96	5.36	1.33	7.49

**Table 8 nanomaterials-14-00641-t008:** Al_x_CoCrFeNi_2.1_ hardness and compression properties of HEAs.

Alloys	Hardness/HV	Yield Strength/MPa	Fracture Strength/MPa	Strain/%
Al_00_	433	1246	1701	39.8
Al_03_	472	1778	1974	28.6
Al_07_	564	2075	2169	20.6
Al_10_	538	1998	2279	19.0
Al_13_	563	2099	2333	15.6

## Data Availability

The data presented in this study are available on request from the corresponding author. The data are not publicly available due to all authors’ decisions.

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
