# Peer review of "Nano-Structure Evolution and Mechanical Properties of AlxCoCrFeNi2.1 (x = 0, 0.3, 0.7, 1.0, 1.3) High-Entropy Alloy Prepared by Mechanical Alloying and Spark Plasma Sintering"

_nanomaterials, 2024, doi:10.3390/nano14070641_

Round 1

Reviewer 1 Report

Comments and Suggestions for Authors

This is an interesting and well presented study where the novelty appears to arise from both the synthesis method and the introduction of Al in the CoCrFeNi system.  The formation of increasing levels of the BCC structure with Al content then prompts the impact on trade-off between fracture strength and fracture strain.  What is perhaps a little surprising is how predictable this relationship is for a high entropy alloy system - as shown in Figure 6)  In general the work is very well described with good experimental detail and well justified conclusions. 

The presentation could be improved as there is a tendency to run a lot of material together - for example the whole of page 4 (from line 111) is a single paragraph which makes it very difficult to digest. It is also not necessary to refer to "Al element" repeatedly in the text where simple Al would do.  

I also struggled with the interpretation of figure 2 where it is difficult to see the evidence of the BCC phase in the XRD patterns, even at the highest Al content.  Although these plots are there to show the time sequence, can the lower traces (longer times) be expanded a little so that the evidence (presumably largely the shoulder on the 110 reflection around 44 deg) to support the conclusions in the text can be seen?  

In Table 4 the use of 4 significant figures in the compositions from EDS cannot be justified but this is also a minor point.  Below Table 5, line 194, there is also no gap to the text - again just a matter of presentation.

Thus the study appears to have been well executed, the paper is well written and the conclusions adequately justified by the results.

Comments on the Quality of English Language

There are relatively few errors in the presentation of the manuscript and minor editing for both spelling and occasionally grammar would help. 

Author Response

The point-by-point response can be seen in attached file.

Reviewer 2 Report

Comments and Suggestions for Authors

The authors of the work “Nano-structure and mechanical properties of AlxCoCrFeNi2.1 (x=0,0.3,0.7,1.0,1.3) high entropy alloy prepared by mechanical alloying and spark plasma sintering”.

They have studied the effect of Al content on the microstructure evolution and mechanical properties of the AlxCoCrFeNi2.1 alloys. I consider that this work is very interesting, but before its publication certain corrections must be made:

1 – It is necessary that a native English speaker to read and correct the entire text, since there are sentences that are difficult to understand.

2 – In section 2 “Experimental details” it is necessary to include a detailed description of the spark plasma sintering conditions: machine model, temperatures, pressure and deformation evolution graph vs. time. In addition, it is necessary to indicate the heating rates and dimensions of the samples obtained.

3 – This sentence is not understood “The compression sample is processed into a cylinder of Φ3mm×6mm by wire cutting, compressive properties at room temperature were measured by an MTS-E45.105 testing machine at experimental rate was static compression 1.0×10- 3S-1.”, correct it.

4 – Include references to the following examples from lines 74-77: including atomic size difference (δ), enthalpy of mixing (∆Hmix), and the combined effect of enthalpy of mixing, entropy of mixing and melting temperature (Ω) of alloy system.

5 - In line 85, correct the sentence and explain what 𝛺𝑖j means.

6 – In line 96 correct the “t” symbol. Also, add a “period” after the word “alloy”.

7 – In line 97 add the abbreviation for “valence electron concentration”.

8 – In line 103 at the end of the sentence add “respectively”.

9 – In Table 1 add a 0 at the Fe-Fe intersection.

10 – In line 107 explain what the abbreviation MA means.

11 – In line 113 change “As we all know” for another construction.

12 – This sentence “Therefore, there are five elements in this system, the XRD characteristic 115 peak of Al element disappears first.” It has no informative value, remove it.

13 - In line 117 include the Figure 2e.

14- In line 123 the phrase “After 20h” is found twice in a row, correct it.

15 – In figures 2 a and b add the symbol for BCC in the legend.

16 – In table 7 correct the VEC column

Author Response

(The authors gave the same response as above.)
